# Methoxyflurane in Non-Life-Threatening Traumatic Pain—A Retrospective Observational Study

**DOI:** 10.3390/healthcare9101360

**Published:** 2021-10-13

**Authors:** Florian Ozainne, Philippe Cottet, Carlos Lojo Rial, Stephan von Düring, Christophe A. Fehlmann

**Affiliations:** 1A.C.E. Genève Ambulances SA, 1225 Geneva, Switzerland; f.ozainne@ace-ambulances.ch (F.O.); p.cottet@ace-ambulances.ch (P.C.); s.vonduring@ace-ambulances.ch (S.v.D.); 2Paramedic School, 1231 Geneva, Switzerland; 3Department of Anaesthesiology, Clinical Pharmacology, Intensive Care and Emergency Medicine, Geneva University Hospitals, 1211 Geneva, Switzerland; 4Emergency Department, Guy’s and St. Thomas’ NHS Foundation Trust, London SE1 9RS, UK; lojorial@hotmail.com; 5Department of Critical Care Medicine, Sunnybrook Health Sciences Centre, Toronto, ON M4N 3M5, Canada; 6School of Epidemiology and Public Health, University of Ottawa, Ottawa, ON K1G 5Z3, Canada; 7Ottawa Hospital Research Institute, Ottawa, ON K1Y 4E9, Canada

**Keywords:** analgesia, prehospital, traumatic, methoxyflurane, pain

## Abstract

Pain management is a key issue in prehospital trauma. In Switzerland, paramedics have a large panel of analgesic options. Methoxyflurane was recently introduced into Switzerland, and the goal of this study was to describe both the effect of this medication and the satisfaction of its use. This was a retrospective cohort study, performed in one emergency ambulance service. It included adult patients with traumatic pain and a self-assessment of 3 or more on the visual analogue scale or verbal numerical rating scale. The primary outcome was the reduction in pain between the start of the care and the arrival at the hospital. Secondary outcomes included successful analgesia and staff satisfaction. From December 2018 to 4 June to October 2020, 263 patients were included in the study. Most patients had a low prehospital severity score. The median pain at arrival on site was 8 and the overall decrease in pain observed was 4.2 (95% CI 3.9–4.5). Regarding secondary outcomes, almost 60% had a successful analgesia, and over 70% of paramedics felt satisfied. This study shows a reduction in pain, following methoxyflurane, similar to outcomes in other countries, as well as the attainment of a satisfactory level of pain reduction, according to paramedics, with the advantage of including patients in their own care.

## 1. Introduction

Acute pain is a frequent symptom in prehospital care. Its prevalence reaches 42%, with 73% of patients receiving analgesics and approximately 50% experiencing an adequate reduction in pain [1,2]. Traumatic pain is usually treated with paracetamol [3], the equimolecular mixture of oxygen and nitrous oxide [4], ketamine [5], and opioids, such as morphine, fentanyl, or sufentanil [6,7,8,9]. In Switzerland, these analgesics can be administered by paramedics without direct physicians’ orders.

Methoxyflurane is a halogenated derivative with strong analgesic properties at sub-anaesthetic doses [10,11]. It was recently approved in Switzerland for the treatment of moderate to severe traumatic pain in conscious adult patients. Under paramedics’ supervision, a patient can self-administer using a 3 mL portable inhaler with a quick onset of action. Convenient, lightweight, compact, single-use, and non-invasive, methoxyflurane has been considered an adjunct therapy to the primary treatment of pain in prehospital care [12].

In the emergency department, trials have shown that methoxyflurane was better than a placebo, without any signs of toxicity [13]. Another recent randomized trial, including some prehospital patients, showed a greater pain reduction at 5 min, compared to standard analgesic treatment [14]. Finally, current prehospital-focused trials being conducted in Italy and Canada should give additional information on the efficacy of methoxyflurane in the prehospital setting [15,16]. No clinical study to date has evaluated the use of methoxyflurane in the Swiss prehospital setting.

The purpose of this study was to describe the characteristics of the patient population that received methoxyflurane for acute traumatic pain in the prehospital setting, its effect on the intensity of pain, and the satisfaction of prehospital care providers.

## 2. Materials and Methods

### 2.1. Study Design and Setting

This was a retrospective observational study reporting the use of methoxyflurane in prehospital care for traumatic pain, in a unique emergency ambulance service (EAS, A.C.E. Genève Ambulances SA, Geneva, Switzerland) in Geneva, Switzerland, between December 2018 and October 2020, based on prospectively collected data. As a result of the burden related to the first COVID-19 wave, the data collection was suspended from 25 March to 4 June 2020.

In Switzerland, the main component of the emergency prehospital response system is an advanced life-support (ALS) ambulance staffed by two paramedics [17]. Ambulance services, public or private, are organized to deliver prehospital care. All missions are coordinated by a state-dependent emergency dispatch center. For acute pain, and without any immediate life-threatening conditions, paramedics have autonomous medication options, without direct medical prescription. Each EAS can have different treatment strategies, depending on the approbation of its medical director (an emergency medicine certified physician) and/or of the local organization. However, options for analgesics used by paramedics are rather standard and include non-opioid agents (such as paracetamol/acetaminophen, nonsteroidal, anti-inflammatory drugs), opioids (morphine, fentanyl, and sufentanil), and psychotropic drugs (ketamine and nitrous oxide). Paramedics from the ambulance service under study have autonomous prescription qualifications for all of these drugs.

In this study, paramedic pain treatment protocols include different steps. The first one is pain assessment with VAS or VNRS. In traumatic cases, paramedics can administer methoxyflurane if the pain is evaluated to be more than 3/10. This administration is performed using a 3 mL inhaler with a concentration of 99.9%. Then, paramedics have different options, depending on pain intensities. If the pain is severe (>7/10) and the patient needs to be moved or if a fracture needs to be realigned, paramedics can use midazolam and ketamine. As soon as the previous painful procedures have been completed and the patient has been scooped, analgesia can be continued by titrating opiates and administering paracetamol. As for any intervention, patients can refuse any treatment they do not want to receive.

### 2.2. Inclusion and Exclusion Criteria

Inclusion criteria were patients 18 years or older with traumatic pain. Exclusion criteria were (1) self-evaluation of the pain not possible, (2) visual analogue scale (VAS) or verbal numerical rating scale (VNRS) ≤ 3, and/or (3) contraindications to methoxyflurane.

### 2.3. Data Collection

Following each intervention, the lead paramedic recorded (in a web-based case report form) all the information regarding the patient and the intervention: age, sex, clinical features, intervention features, treatment administered, and satisfaction of the paramedic team. Clinical features included pain localization, systolic and diastolic blood pressure, heart rate, and oxygen saturation at arrival on site and upon arrival at the hospital’s ED. Treatment administrated was categorized in four groups: methoxyflurane alone, methoxyflurane combined without opioids, methoxyflurane combined with opioids only, and methoxyflurane combined with a mix of opioids and non-opioids. Intervention features included timing (night (defined as between 7:00 PM and 7:00 AM) and weekends), location (home, public area, work/school/sport/leisure activity, and others), and the severity of the emergency using the National Advisory Committee for Aeronautics (NACA) score. Pain intensity was measured using the VAS or VNRS [18]. These scales are the ones usually used by paramedics and are among the most frequently used for pain evaluation. For the VAS, the patients are asked to rank their pain from 0 (no pain) to 10 (worst imaginable pain). For the VNRS, a plastic rule is used, and the patient can move a slider based on pictograms. On the opposite side, graduation allows paramedics to assess pain intensity from 0 to 10. Pain was assessed directly with the patients and recorded at four specific intervention times: on arrival (T0), at scooping time (T1), in the ambulance (T2), and at hospital delivery (T3). Fixed times (5, 10, and 20 min) were deliberately not used for practical reasons, as they do not reflect current practices in the local setting. Finally, the satisfaction of the paramedic leader was reported using a five-point Likert scale. Data from the intervention report was automatically extracted from the data base. Then, one author (CF) oversaw the cleaning of the data set and analyses.

### 2.4. Outcomes

The primary outcome was the decrease in VAS from T0 (paramedics’ arrival on site) to T3 (hospital delivery). Secondary outcomes were a successful analgesia (defined as a T3 VAS or VNRS ≤ 3) and an overall good paramedic satisfaction level (defined as satisfied or very satisfied).

### 2.5. Statistical Analysis

Demographic data are expressed as means ± standard deviation or frequency and proportion, based on the type of variables. As the main goal was a descriptive analysis, no sample size calculation was performed prior to this study. For all the tests, a two-sided *p*-value below 0.05 was considered significant.

### 2.6. Research Ethics Review

As this study was considered a quality improvement project on previously collected data, the ethics approval was waived by the institutional review board.

## 3. Results

From 13 December 2018 to 25 March 2020 and from 4 June to 14 October 2020, paramedics responded to 7120 calls, of which 1866 involved patients with trauma. Of these, 35% of them required pain management. After the exclusion of 378 patients (self-evaluation of the pain impossible, VAS or VNRS ≤ 3, or contraindications to methoxyflurane), 263 patients were included (Figure 1).

Table 1 presents the baseline characteristics of the patients. The average age was 54 (SD = 22), with an even sex ratio (53% versus 47%). Above 90% of patients had a low prehospital severity score (NACA 3 or less). Almost half of the interventions took place in public places, and about 30% were in patient’s homes. The pain localisation was mostly in the limbs (45.3% for lower limbs and 28.9% for upper limbs), followed by the back (14.1%) and thorax (7.6%). All patients had normal vital signs, and none had an imminent life-threatening condition requiring immediate intervention.

The median VAS or VNRS on arrival on site was 8 (IQR 6–10). Methoxyflurane was given alone in 35.4% of patients, in combination with non-opioid medication in 11.3%, with opioids only in 12.5%, and with a mixed combination to the remaining 41.1%.

Table 2 presents the results of the different outcomes. For the primary outcome, the evolution of the pain over time is presented in Figure 2. Overall, the average decrease in VAS or VNRS was 4.2 (95% CI 3.9–4.5) on a scale of 10. The subgroup analysis demonstrated a 3.7 reduction with methoxyflurane alone, a 4.7 reduction when combined with non-opioid medication, a 3.8 reduction when combined with opioids only, and a 4.6 reduction with a mixed combination (Table 2). Regarding the secondary outcome for pain, 57.8% (152/263) had a successful analgesia (defined as a T3 VAS or VNRS ≤ 3), which increased to 74.5% (196/263) when adjusting for patients refusing additional analgesia.

Overall, the paramedic satisfaction was good, with 30.0% of paramedics satisfied and 41.4% of them very satisfied with methoxyflurane pain management.

## 4. Discussion

This retrospective cohort study demonstrated that methoxyflurane appears to participate in pain reduction following trauma. It also showed that paramedics were satisfied or very satisfied in the majority of cases with the use of this medication.

To the authors knowledge, this is the first study reporting the use of methoxyflurane in a Swiss prehospital setting. Based on a larger number of patients, it showed an average of 4.2-point decrease in pain after the use of methoxyflurane. While other causes, such as anxiety reduction, could explain these results, this study demonstrated a greater improvement, compared with other recent studies which looked at intra-hospital analgesia [13]. The recent MEDITA trial, which included prehospital patients, exhibited a similar improvement in pain to the present results [14]. Other prehospital-based studies showed improvement between 2 and 3.5 points, lower than the overall decrease observed in this study [19,20,21]. Interestingly, the analysis of the methoxyflurane-alone group showed that the improvement was still higher than most of these studies. However, more than 60% of the patients required another medication. This suggests that a polyanalgesic approach to pain is necessary to achieve good analgesic effect, and that methoxyflurane should be considered an additional resource, rather than a silver bullet.

Oligo-analgesia, which is defined as failure to provide sufficient analgesia, is an important issue in prehospital care [22]. In previous studies (also conducted in Switzerland), the prevalence of successful analgesia was around 57% and 62%, similar to this study [23,24]. This proportion is even higher when adjusting for patients who evaluated their pain at 4 or more but said, at the same time, that they were comfortable enough and did not want additional pain medication. Finally, a recent, large observational study of the clinical practice of prehospital analgesia in Zurich, Switzerland, demonstrated a similar result to us, with 77% of sufficient analgesia, but without using methoxyflurane [25]. This reinforces the authors’ opinion that methoxyflurane is a useful medication but should be used in combination with other traditional drugs.

The present study demonstrated an overall good level of satisfaction. This could be explained by the simplicity and usability of the device, especially in situations where security equipment cannot be immediately removed, thus compromising iv access. Buntine et al. showed a higher satisfaction than this study did (over 80%) [19]. This can be explained by differences in the study population.

Importantly, although data was prospectively collected, it was a retrospective study with only one EAS included. Nevertheless, these results should match other, similar EAS in Switzerland, as the patients and the paramedic training are comparable. Secondly, there was no comparator. The goal of this study was to describe characteristics of the patients, as well as the evolution of their pain and satisfaction. Further comparative studies should be performed to ascertain the efficiency and efficacy of methoxyflurane, compared to other analgesics in prehospital setting. Another limitation is the fact that only patients with traumatic pain were included. While recent studies showed that methoxyflurane could also be used in non-traumatic pain, such as renal colic pain or the exacerbation of chronic back pain [26], it was not possible to include these patients in our study because of the restriction in the licenced use in Switzerland. Finally, the study was briefly suspended during the first COVID-19 wave, as the workload was too heavy to allow paramedics to collect data. Although this should not affect the validity of the results, data collection during the weeks of the suspension of the study could have been affected.

To summarize, this study shows similar results in pain reduction following methoxyflurane in a Swiss cohort as in other countries, with a good paramedics satisfaction. This new medication is, therefore, an additional tool that will contribute to the continuous improvement of pain management, especially in the prehospital setting, where proper training and delegated medical interventions to paramedics has greatly improved care these last 20 years.

## 5. Conclusions

Methoxyflurane seems to offer a similar reduction in pain in Switzerland as in other countries and a good paramedic satisfaction, with the advantage of including the patients in their care.

## Figures and Tables

**Figure 1 healthcare-09-01360-f001:**
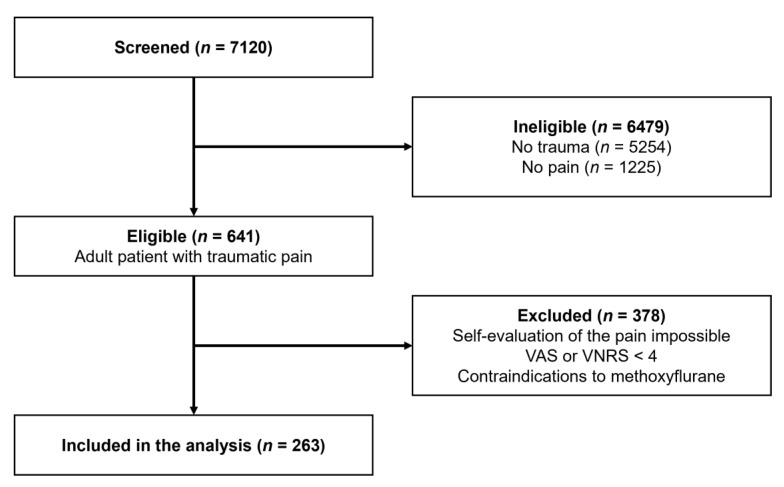
Flowchart.

**Figure 2 healthcare-09-01360-f002:**
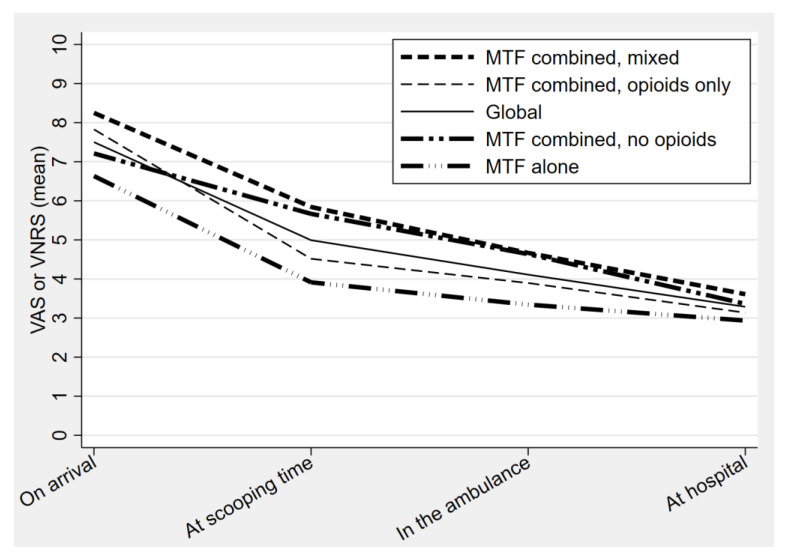
Evolution of the pain over the time (by group of treatment).

**Table 1 healthcare-09-01360-t001:** Patients characteristics.

Variables	All Patients (*n* = 263)
Sex (female)—*n* (%)	123 (46.8)
Age (years)—mean ± SD	53.6 ± 22.5
Weekend intervention—*n* (%)	55 (20.9)
Night intervention—*n* (%)	70 (26.6)
Place of intervention—*n* (%)	
Public place	121 (46.0)
Home	85 (32.3)
Work/school/sport/leisure	46 (17.5)
Others	11 (4.2)
Pain localisation—*n* (%)	
Lower limb	119 (45.3)
Upper limb	76 (28.9)
Back	37 (14.1)
Thorax	20 (7.6)
Pelvis	6 (2.3)
Skull	5 (1.9)
Systolic blood pressure (mmHg)—mean ± SD	137 ± 24
Diastolic blood pressure (mmHg)—mean ± SD	82 ± 15
Heart rate (/min)—mean ± SD	81 ± 14
Oxygen saturation (%)—mean ± SD	97 ± 2
Initial VAS or VNRS score—median (IQR)	7 (6–10)
Treatment—*n* (%)	
Methoxyflurane alone	93 (35.4)
Methoxyflurane combined (no opioids)	29 (11.3)
Methoxyflurane combined (opioids only)	33 (12.5)
Methoxyflurane combined (mixed)	108 (41.1)

**Table 2 healthcare-09-01360-t002:** Pain outcomes.

Outcome	All Patients (*n* = 263)	MTF ^1^ Alone (*n* = 93)	MTF ^1^ Combined, No Opioids (*n* = 29)	MTF ^1^ Combined, Opioids Only (*n* = 33)	MTF ^1^ Combined, Mixed (*n* = 108)
Pain improvement—mean (95% CI)	4.2 (3.9–4.5)	3.7 (3.2–4.2)	4.7 (3.8–5.6)	3.8 (2.6–5.1)	4.6 (4.2–5.1)
Successful analgesia—*n* (%)	152 (57.8)	55 (59.1)	20 (69.0)	20 (60.6)	57 (52.8)
Modified successful analgesia ^2^—*n* (%)	196 (74.5)	77 (82.8)	23 (79.3)	25 (75.8)	71 (65.7)

^1^ MTF = Methoxyflurane; ^2^ VAS or NVRS ≤ 3 or refusing additional treatment.

## Data Availability

The data and the Stata code for this study are freely available on the Open Science Framework (https://doi.org/10.17605/OSF.IO/9M7FU (accessed on 12 October 2021)).

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
