# Peer review of "Methoxyflurane in Non-Life-Threatening Traumatic Pain—A Retrospective Observational Study"

_healthcare, 2021, doi:10.3390/healthcare9101360_

Round 1

Reviewer 1 Report

Patients who received Methoxyflurane experienced a reduction in pain, and the authors concluded that Methoxyflurane is useful in reducing pain. However, as there were no results for the non-Methoxyflurane group, there was no evidence from anywhere that Methoxyfluranewas any more useful than non-Methoxyflurane. Therefore, it seems hard to prove the efficacy of Methoxyflurane for pain relief from the present results.

Assessment methods of Visual Analogue Scale (VAS) or verbal numerical rating scale (VNRS) are obscure.

Pain in patients with trauma generally decreases as anxiety is reduced. How did the authors evaluate this?

How much and when was Methoxyflurane administered to subjects and by which route?

Author Response

Thank you very much for your review. You will find our comments and our modifications in the attachment. 

Reviewer 2 Report

This paper addresses the controversial research field,  namely pain management. Although the study might be significant for the clinical practice, in its current form the article is ready for publication. I suggest that the authors revise the paper from structure to English and spelling error up to the importance of the study and the added value of their findings. A more adequate state of the art is needed (there is recent research on pain management, 10.1109/TBME.2018.2854917 10.1016/j.jare.2021.04.004 10.1109/ACCESS.2021.3049880 ). Also, the authors should put more accent on the outcomes of this study, which at this stage is very briefly described. In terms of statistical analysis, a more detailed analysis is required. Moreover, the authors should include a section where to describe the protocol of the study. The results section is also quite poor. More details regarding the analysis performed to draw the conclusions claimed in section 4 are necessary. The conclusion section is unacceptable, the 2 lines text does not reflect the aim of the study and the outcome of the study. 

Thus, the authors have to rigorously revise the paper before being considered for publication.  

Author Response

(The authors gave the same response as above.)

Round 2

Reviewer 1 Report

Methods;
The details of assessing a VAS score need to be described. In the present manuscript, it is unclear what a high score indicates.

Result& discussion
The authors have been able to reach consistent results by revising the text.
In the previous version of the article, the authors claimed that methoxyflurane was useful in reducing VAS scores. In the revised version, the authors only speculated on the comparison with previous reports. The conclusion seems to be acceptable.
Since the main discussion has been switched to a comparison with other studies, it is desirable to prepare a table comparing this study with past studies.

Author Response

1) We considered that pain assessment in prehospital is something usually widely known. However, folllowing your comment, we added some precisions: "For the VAS, the patients are asked to rank their pain from 0 (no pain) to 10 (wosrt imaginable pain). For the VNRS, a plastic rule is used, and the patient can move a slider based on pictograms. On the opposite side, graduation allows paramedics to assess pain intensity from 0 to 10. "

2) We consider that such a table is currently not possible. First, such a table should be the result of a systematic research, based on multiple databases, as recommended by PRISMA guidelines. This would be a study design per se. Second, it seems impossible to perform such a table in a reasonable delay. Finally, a table that would summarize only the studies we cite would result in biases, in a direction or another, based on the absence of systematic research. We hope that, while we were not able to follow your proposition, you would consider this manuscript for publication.